

# De novo oviduct transcriptome of the moor frog *Rana arvalis*: a quest for maternal effect candidate genes

Longfei Shu[1,2], Jie Qiu[3] and Katja Räsänen[1,2]

[1] Department of Aquatic Ecology, Eawag, Swiss Federal Institute of Aquatic Science and Technology, Dübendorf, Switzerland
[2] Institute of Integrative Biology, ETH Zürich, Swiss Federal Institute of Technology in Zürich, Zürich, Switzerland
[3] Institutue of Crop Science and Institute of Bioinformatics, College of Agriculture and Biotechnology, Zhejiang University, Hangzhou, China

## ABSTRACT

Maternal effects can substantially affect ecological and evolutionary processes in natural populations. However, as they often are environmentally induced, establishing their genetic basis is challenging. One important, but largely neglected, source of maternal effects are egg coats (i.e., the maternally derived extracellular matrix that surrounds the embryo). In the moor frog, the gelatinous egg coats (i.e., egg jelly) are produced in the mother's oviduct and consist primarily of highly glycosylated mucin type O-glycans. These O-glycans affect jelly water balance and, subsequently, contribute to adaptive divergence in embryonic acid tolerance. To identify candidate genes for maternal effects, we conducted RNAseq transcriptomics on oviduct samples from seven *R. arvalis* females, representing the full range of within and among population variation in embryonic acid stress tolerance across our study populations. *De novo* sequencing of these oviduct transcriptomes detected 124,071 unigenes and functional annotation analyses identified a total of 57,839 unigenes, of which several identified genes likely code for variation in egg jelly coats. These belonged to two main groups: mucin type core protein genes and five different types of glycosylation genes. We further predict 26,711 gene-linked microsatellite (simple sequence repeats) and 231,274 single nucleotide polymorphisms. Our study provides the first set of genomic resources for *R. arvalis*, an emerging model system for the study of ecology and evolution in natural populations, and gives insight into the genetic architecture of egg coat mediated maternal effects.

## BACKGROUND

Understanding evolutionary processes of natural populations necessitates a good understanding of the genetic architecture of trait variation in an ecologically relevant context (*Houle, Govindaraju & Omholt, 2010*; *Mitchell-Olds, Willis & Goldstein, 2007*; *Nadeau & Jiggins, 2010*). Maternal effects (the effects of a mother's environment

Corresponding author
Longfei Shu, longfei.shu@wustl.edu

and phenotype on offspring performance) are an important source of phenotypic variation and often under strong natural selection (reviewed in, *Mousseau & Fox, 1998*; *Räsänen & Kruuk, 2007*). Maternal effects can influence speed and direction of evolution as well as facilitate local adaptation (*Hangartner, Laurila & Räsänen, 2012*; *Räsänen & Kruuk, 2007*; *Shu et al., 2016*; *Wolf et al., 1998*). However, as maternal effects are typically at least partially environmentally induced, their genetic architecture is poorly understood.

Maternal effects can arise through various mechanisms, most commonly acknowledged through variation in egg size and content (reviewed in, *Bernardo, 1996*; *Mousseau & Fox, 1998*). A much less well acknowledged but important source of maternal effects are so called egg coats (*Shu, Suter & Räsänen, 2015b*). Egg coats (present in all sexually reproducing animals, as well as many asexual metazoans) are maternally derived, extracellular structures that consist of multiple functionally and structurally different layers (reviewed in, *Menkhorst & Selwood, 2008*; *Shu, Suter & Räsänen, 2015b*; *Wong & Wessel, 2006*). These structures have many key functions, ranging from fertilization to embryonic protection (*Shu, Suter & Räsänen, 2015b*). The genetic basis of the innermost, so called oocyte coats, has been studied in several model systems, such as sea urchins and the abalone (*Claw & Swanson, 2012*; *Palumbi, 2009*), the moth *Bombyx mori* (*Papantonis, Swevers & Iatrou, 2015*) and the frog *Xenopus laevis* (*Hedrick, 2008*; *Shu, Suter & Räsänen, 2015b*). However, the genetic basis of the outermost gelatinous egg coats, often found in taxa where embryos develop externally in the surrounding environment (e.g., various invertebrates, fish, and amphibians), is to date effectively unstudied (reviewed in, *Shu, Suter & Räsänen, 2015b*).

In the moor frog *Rana arvalis*, jelly coat mediated maternal effects have driven adaptive divergence in embryonic acid stress tolerance (*Hangartner, Laurila & Räsänen, 2011*; *Persson et al., 2007*; *Räsänen, Laurila & Merilä, 2003b*; *Shu, Laurila & Räsänen, 2015a*; *Shu et al., 2015c, 2016*). This divergence is primarily due to glycoproteins of the jelly coats, which influence water balance and, subsequently, embryonic survival in acidic conditions (*Shu, Laurila & Räsänen, 2015a*). Egg jelly glycoproteins are complex glycan structures attached to a protein backbone (*Hedrick & Nishihara, 1991*) and are highly species specific (*Coppin et al., 1999*; *Delplace et al., 2002*; *Strecker et al., 2003*). Given this complexity, the jelly coats can be coded by multiple genes that regulate, for instance, the protein backbone and the activities of enzymes that impact how different glycan branches are attached (*Shu, Suter & Räsänen, 2015b*). This makes identifying the genetic basis of jelly coat variability highly challenging—particularly so in natural populations not amenable to experimental cross-generational rearing or genetic manipulations. Here next generation sequencing tools, in particular RNA-seq, can be helpful (*Todd, Black & Gemmell, 2016*; *Wang, Gerstein & Snyder, 2009*). Here, we applied de novo transcriptomics to get first insight to jelly coat genes in *R. arvalis*. We conducted tissue-specific RNA-seq of *R. arvalis* oviducts, where jelly coat biosynthesis takes place (*Hedrick & Nishihara, 1991*).

Despite the increasing numbers of genomes being sequenced, the genomic resources for amphibians are sparse and only a handful of amphibian genomes have been sequenced

so far (*Elewa et al., 2017*; *Hammond et al., 2017*; *Hellsten et al., 2010*; *Nowoshilow et al., 2018*; *Sun et al., 2015*). This is an important shortcoming, given that amphibians are common model systems for a range of ecotoxicological (*Helbing, 2012*), ecological and evolutionary studies, the latter ranging from spatial patterns of phenotypic and genetic divergence (e.g. *Egea-Serrano et al., 2014*; *Richter-Boix et al., 2011*; *Van Buskirk & Arioli, 2005*) to developmental plasticity (*Gomez-Mestre, Touchon & Warkentin, 2006*) and adaptive maternal effects (*Mousseau & Fox, 1998*; *Räsänen, Laurila & Merilä, 2003b*; *Shu et al., 2015c*, *2016*).

To bridge this knowledge gap—and as a first step to identify candidate genes for egg coat mediated maternal effects—we here apply tissue-specific transcriptomics on *R. arvalis* oviducts. In doing so, we complement recent transcriptomics studies on other ranid species (*Birol et al., 2015*; *Helbing, 2012*; *Price et al., 2015*; *Qiao et al., 2013*; *Robertson & Cornman, 2014*; *Zhang et al., 2013*) and increase availability of genomics resources for amphibians in general. As our specific interest is in identifying putative genes underlying egg jelly coat variation, we collected samples from the oviduct on seven *R. arvalis* individuals encompassing the full range of embryonic acid tolerance variation among and within our study populations (*Hangartner, Laurila & Räsänen, 2011*; *Shu, Laurila & Räsänen, 2015a*; *Shu et al., 2015c*). We focused particularly on identifying genes related to biosynthesis of mucin type O-linked glycans. In addition, this study provides the first transcriptomes for *R. arvalis*, and adds more genome wide markers in addition to existing resources (*Brelsford et al., 2017*).

## MATERIALS AND METHODS

### Study system

*Rana arvalis* is an anuran amphibian, broadly distributed across the western Palearctic (*Glandt, 2006*). Individuals from three *R. arvalis* populations (Tottajärn, [T], Bergsjö, [B], and Stubberud, [S]) breeding in permanent ponds along an acidification gradient in southwestern Sweden were used in this study (*Hangartner, Laurila & Räsänen, 2011*; *Shu, Laurila & Räsänen, 2015a*; *Shu et al., 2015c*, *2016*; *Shu, Suter & Räsänen, 2015b*). These ponds differ in pH due to a mix of acid rain, natural acidification processes, and variation in bedrock buffering capacity (*Räsänen, Laurila & Merilä, 2003a*, *2003b*). These populations were chosen because, based on former common garden experiments (*Hangartner, Laurila & Räsänen, 2012*; *Shu et al., 2015c*), they represent the extreme ends of adaptive divergence in embryonic acid stress tolerance (i.e., [S] most acid sensitive; [B] intermediate tolerance; [T] most acid tolerant) across our study gradient. The pH in these ponds ranges from highly acidic (pH 4, site [T]) to intermediate (pH 6, site [B]) to neutral (pH 7.5, site [S]).

During the breeding season of 2012, females in breeding condition were collected and transported to the laboratory at Uppsala University (59°50′N, 17°50′E). The females were maintained in containers with moist *Sphagnum* moss (antibacterial medium) in a climate chamber at 2–4 °C until artificial crosses (see below) and RNA sampling were conducted a few days later. The females used for this transcriptomics study consisted of a subset (out of 7–10 females/population) of females that were artificially crossed to establish acid tolerance of embryos in a common garden laboratory experiment and to study

variation in macromolecular composition of egg jelly coats (for details see, *Shu et al., 2015c*, *2016*). The experiments were conducted under permissions from the Västra Götaland county board (collection permit: Dnr 522-6666-2011) and the Ethical committee (Dnr C65/11) for animal experiments in Uppsala County.

## Sampling and RNA extraction

To bracket a broad range of genetic backgrounds, a total of six females were chosen so as to represent females that produced the most acid tolerant (highest embryonic survival at acidic pH in a common garden experiment) and most acid sensitive (lowest embryonic survival at acidic pH in a common garden experiment) clutches both among and within each of the three populations (*Shu et al., 2016*). In addition, we included a sample from one female that had not yet fully ovulated, hence providing a reference for gene expression at an earlier stage of egg coat production (Supplementary Material). For each female, RNA of the whole oviduct (i.e., specific tissue where egg jelly is produced; *Hedrick & Nishihara, 1991*) was collected and stored in RNA later at −20 °C until extraction. RNA extraction was conducted using TRIzol (Life Technologies, Basel, Switzerland) according to the manufacturer's protocol, followed by DNase (Qiagen, Venlo, Netherlands) treatment to eliminate potential genomic DNA contamination. Both the concentration and integrity of the RNA samples for transcriptomic analyses were evaluated with the Agilent 2100 Bioanalyzer. All samples had an RNA integrity value (RIN) >8 and were, hence, used to construct the cDNA libraries (*Schroeder et al., 2006*).

## cDNA library construction and sequencing

cDNA libraries were generated using the TruSeq RNA-Seq Sample Prep kit according to the manufacturer's protocol (Illumina Inc., San Diego, CA, USA). Briefly, magnetic beads with Oligo (dT) were used to isolate the poly(A) + mRNA. Fragmentation buffer was added in the presence of divalent cations to break the mRNA into short fragments of approximately 200 bp. Short fragments were purified with the QiaQuick PCR extraction kit, followed by end reparation, adding poly(A) and sequencing adapters. The suitable fragments were selected for the PCR amplification as templates. In total, seven cDNA libraries were constructed and sequenced using the Illumina HiSeq$^{TM}$ 2000 (90 bp paired-end reads). The sequencing reactions were conducted at the Beijing Genomics Institute (BGI), Shenzhen.

## Assembly and annotation

Raw reads were filtered to remove reads containing adaptors, reads with unknown nucleotides greater than 5%, and low-quality reads with more than 20% bases of quality value ≤10 (filter_fq, BGI). Only clean reads were used in the following analyses. Transcriptome *de novo* assembly was carried out using the assembly program Trinity (*Grabherr et al., 2011*). Briefly, the software first combined reads of certain lengths of overlap to form contigs. Subsequently, the reads were mapped back to the contigs, which were connected until extension proceeded on neither end (*Grabherr et al., 2011*). After removing any redundancy, the resulting sequences were defined as unigenes. Software and their

parameters were: Trinity (–seqType fq –min_contig_length 100; –min_glue 4 –group_pairs_distance 250; –path_reinforcement_distance 85 –min_kmer_cov 4); TGICL (-l 40 -c 10 -v 20); Phrap (-repeat_stringency 0.95 -minmatch 35 -minscore 35).

Unigene annotation provides information of expression and functional annotation of unigene sequences. Unigene sequences were first annotated using blastx against the database Nr (http://www.ncbi.nlm.nih.gov/), with a cut-off $E$-value of 1$e$-5. To acquire a more comprehensive annotation, unigene sequences were also aligned to the protein databases Swiss-Prot, Kyoto Encyclopedia of Genes and Genomes (KEGG) and Gene Ontology (GO) (1$e$-5) by blastx. In order to predict and classify the possible functions of the unigenes, they were additionally annotated to Cluster of Orthologous Group (COG), which classifies orthologous gene products (*Tatusov et al., 2003*). Software and their parameters were: BLAST (-F F -e 1e-5 -p blastn; -F F -e 1e-5 -p blastx); Blast2GO (Default); Path_finder (Default).

## Protein coding sequence prediction

Unigenes were aligned by blastx (*e* value < 0.00001) to protein databases in the priority order of NR, Swiss-Prot, KEGG, and COG. Unigenes aligned to a higher priority database were not aligned to lower priority database. Proteins with the highest ranks in the blast results were used to decide the coding region sequences of the unigenes. The coding region sequences were subsequently translated into amino acid sequences with the standard codon table. This produced both the nucleotide sequences (5′ to 3′) and amino acid sequences of the unigene coding regions. Unigenes that could not be aligned to any database were scanned by ESTScan (*Iseli, Jongeneel & Bucher, 1999*), producing nucleotide sequence (5′ to 3′) direction and amino acid sequence of the predicted coding region.

## Metabolic pathway analysis

To investigate the functions of the unigenes in metabolic processes, we acquired pathway annotation by mapping the unigenes to GO and KEGG database (*Kanehisa et al., 2008*). GO database is a relational database, which has three ontologies: molecular function, cellular component, and biological process. KEGG is a database for understanding high-level functions and utilities of the biological system. It is able to analyze gene products during metabolism and cellular processes and allowed us to get pathway annotation for unigenes. We used the Blast2GO program with default setting to acquire GO and KEGG annotation of Unigenes (*Tarazona et al., 2011*).

## Identification of candidate genes for maternal effects

Based on prior work, jelly coats of *R. arvalis* and other amphibians (*Coppin et al., 1999*; *Guerardel et al., 2000*; *Strecker et al., 2003*) mostly consist of mucin type O-glycans, making genes related to mucins and to O-glycan biosynthesis particularly relevant as maternal effect genes. Therefore, we considered as candidate genes related to *R. arvalis* egg jelly coats (*Shu, Suter & Räsänen, 2015b*) unigenes that (i) were expressed in all individuals, (ii) were mapped to the category of "extracellular matrix" (ECM) in GO and KEGG annotations, and (iii) were involved in the glycosylation process.

## Preliminary differential expression analysis

Unigene expression was calculated using the FPKM (fragments per kb per million reads), which can eliminate the influence of different gene length and sequencing level on the calculation of gene expression (*Mortazavi et al., 2008*). Group comparison between populations was performed using the R Bioconductor package NOISeq, which is a data-adaptive and nonparametric method (*Tarazona et al., 2011*). We defined $p \geq 0.8$ and the absolute value of Log2Ratio $\geq 1$ as the threshold to judge the significance of gene expression difference between populations. KEGG enrichment analysis was performed with a Fisher's exact test in Blast2GO (*Tarazona et al., 2011*). Pathway enrichment analysis identifies significantly enriched metabolic pathways or signal transduction pathways in differentially expressed genes (DEGs). After correction for multiple testing, we chose pathways with $Q$ value $\leq 0.05$ as significantly enriched in DEGs (*Gotz et al., 2008*; *Tarazona et al., 2011*).

## Genetic marker resources

To aid population genomic analyses for taxa for which no genome is available, RNA-seq data can be used to identify genomic markers, such as simple sequence repeats (SSRs) and single nucleotide polymorphisms (SNPs), in transcribed regions (*Lopez-Maestre et al., 2016*). We therefore used the oviduct RNA-seq data to predict SSR and SNP markers, which allows later validation of targeted questions (*Lopez-Maestre et al., 2016*), such as allelic frequency variation of the candidate genes in the source populations.

Simple sequence repeats were identified in the final assembly with the software MicroSAtellite (MISA, http://pgrc.ipk-gatersleben.de/misa/) using unigenes as the reference. Assembled contigs were scanned for SNPs with SNP detection software SOAPsnp (*Li et al., 2008*). The program can assemble consensus sequence for the genome of a newly sequenced individual based on the alignment of the raw sequencing reads on the unigenes. The program calculated the likelihood of each genotype at each site based on the alignment of short reads to a unigene, set together with the corresponding sequencing quality scores. It then infers the genotype with highest posterior probability at each site based on Bayes' theorem (the reverse probability model; *Li et al., 2008*). Therefore, the intrinsic bias or errors that are common in Illumina sequencing data have been taken into account and the quality values for use in inferring consensus sequence have been recalibrated. Software and their parameters were: SOAPsnp (Release 1.03, http://soap.genomics.org.cn/soapsnp.html, -u t -Q i -L 90).

## Additional analyses

We performed post hoc analyses to address three additional questions. To test whether embryonic acid tolerance is related to the transcriptional activity, we performed a correlation analysis between the acid tolerance of each individual and the expression of each unigene (FPKM) by PYTHON. A strict correlation threshold was applied (Pearson correlation, $|r| > 0.9$ and $p$-value $< 0.01$).

As our analyses matched partially a fluke (*Clonororchis sinensis*) genome (see results), we tested how much of the fluke genomes our *R. arvalis* dataset covers, we downloaded the genomes of three fluke species (*Clonorchis sinensis, Schistosoma mansoni, S. japonicum*)

**Table 1 Results of RNA sequencing of six *R. arvalis* oviducts.**

| Sample ID | Total raw reads | Total clean reads | Total clean nucleotides | Q20 percentage (%) | GC percentage (%) |
|---|---|---|---|---|---|
| S1 | 99,177,178 | 86,475,930 | 7,782,833,700 | 98.09 | 44.68 |
| S2 | 99,884,602 | 87,485,924 | 7,873,733,160 | 98.04 | 45.11 |
| B1 | 92,353,906 | 81,166,804 | 7,305,012,360 | 98.07 | 44.92 |
| B2 | 96,108,078 | 85,373,136 | 7,683,582,240 | 98.05 | 44.83 |
| T1 | 95,886,666 | 84,428,864 | 7,598,597,760 | 98.10 | 44.74 |
| T2 | 108,026,198 | 84,548,126 | 7,609,331,340 | 98.27 | 45.68 |

**Notes:**

Total reads and total nucleotides are given after adaptor trimming and quality filtering. Q20 percentage is the proportion of nucleotides with a quality value larger than 20; GC percentage is the proportion of guanidine and cytosine nucleotides among total nucleotides. The sample ID indicates the six different females, originating from three populations (S, neutral origin; T, acid origin; and B, intermediate pH origin). The individuals were chosen so as to maximize variation in embryonic acid tolerance (which in turn is largely determined by the molecular composition of the egg jelly coats). In each population, individual 1 (italics) represents a female whose offspring was most acid sensitive in the embryonic stage, while individual 2 represents a female whose offspring was the most acid tolerant (based on screening of embryonic acid tolerance in a laboratory experiment, *Shu et al., 2016*).

from the wormbase (https://parasite.wormbase.org/), and blasted each species (BLATP, *E*-value < 1*e*-7) against our unigenes dataset.

To examine whether there are any hits to the transcriptomic resources developed for the green frog *Rana clamitans* and chorus frog *Pseudacris regilla* (*Robertson & Cornman, 2014*), we downloaded their assembled transcriptomes from NCBI BioProjects (PRJNA162931 and PRJNA163143), and blasted each species against our unigenes database (BLASTX, *E*-value < 1*e*-7).

# RESULTS

## Sequencing and assembly

In total, 53,330,025,420 bp bases were generated from the *R. arvalis* oviduct transcriptome. Total clean reads of the seven cDNA libraries ranged from 81,166,804 to 87,485,924 (Table 1), with an average GC content of 44.95%. In the final assembly, 87,401–112,136 unigenes were detected across the six cDNA libraries (Table 2). Interestingly, for the one female that had not yet ovulated fewer unigenes (69,987) were detected than for the remaining females (that had ovulated). When the cDNA libraries were pooled, a total of 124,071 unigenes were detected, with an N50 of 1,212 bp and a total length of 90.3 Mb. The average length of the unigenes was 728 bp (Fig. 1).

## Transcriptome annotation

The *E*-value distributions of the unigenes showed that approximately 60% of the unigenes had very strong homogeny (<1*e*-30) with the Nr database, while the rest ranged from 1*e*-5 to 1*e*-30 (Fig. 2A). A total of 30% of the unigene sequences had over 80% similarity with the Nr database, while the similarity of the remaining 70% of sequences ranged from 17% to 80% (Fig. 2B). The *R. arvalis* sequences matched best with *Xenopus (Silurana) tropicalis* (43.4%), followed by *X. laevis* (13.1%, Fig. 2C) and the liver fluke *C. sinensis* (12.1%, Fig. 2C). In addition, we found that the percentages of genes in three fluke genomes that have hits in our dataset were very high (*C. sinensis*: 61.4%;

**Table 2 Contigs and unigenes in the transcriptome assembly of six *R. arvalis* oviducts.**

| | Sample ID | Total number | Total length (nt) | Mean length (nt) | N50 | Total consensus sequences | Distinct clusters | Distinct singletons |
|---|---|---|---|---|---|---|---|---|
| Contig | *S1* | *194,272* | *53,890,758* | *277* | *362* | – | – | – |
| | S2 | 217,356 | 59,657,739 | 274 | 366 | – | – | – |
| | *B1* | *174,223* | *49,240,955* | *283* | *392* | – | – | – |
| | B2 | 183,977 | 51,106,261 | 278 | 381 | – | – | – |
| | *T1* | *160,756* | *43,939,551* | *273* | *342* | – | – | – |
| | T2 | 153,751 | 46,501,333 | 302 | 436 | – | – | – |
| Unigene | *S1* | *104,705* | *50,126,396* | *479* | *693* | *104,705* | *15,515* | *89,190* |
| | S2 | 112,136 | 55,372,134 | 494 | 729 | 112,136 | 16,008 | 96,128 |
| | *B1* | *90,424* | *46,895,863* | *519* | *830* | *90,424* | *13,740* | *76,684* |
| | B2 | 91,647 | 47,344,659 | 517 | 787 | 91,647 | 14,419 | 77,228 |
| | *T1* | *91,528* | *40,874,115* | *447* | *632* | *91,528* | *11,288* | *80,240* |
| | T2 | 87,401 | 45,945,512 | 526 | 839 | 87,401 | 12,775 | 74,626 |
| | All | 124,071 | 90,322,330 | 728 | 1,212 | 124,071 | 28,452 | 95,619 |

**Notes:**
The sample ID indicates the six different females, originating from three populations (S, neutral origin, T, acid origin, and B, intermediate pH origin) with the acid most sensitive within each population indicated in italics (See Table 1 for details). N50 is the shortest sequence length at 50% of the transcriptome. Total consensus sequences represents all the assembled unigenes. Distinct Clusters represents the cluster unigenes. The same cluster contains some highly similar (more than 70%) unigenes, and these unigenes may come from the same gene or a homologous gene. Distinct singletons represents that these unigenes come from a single gene.

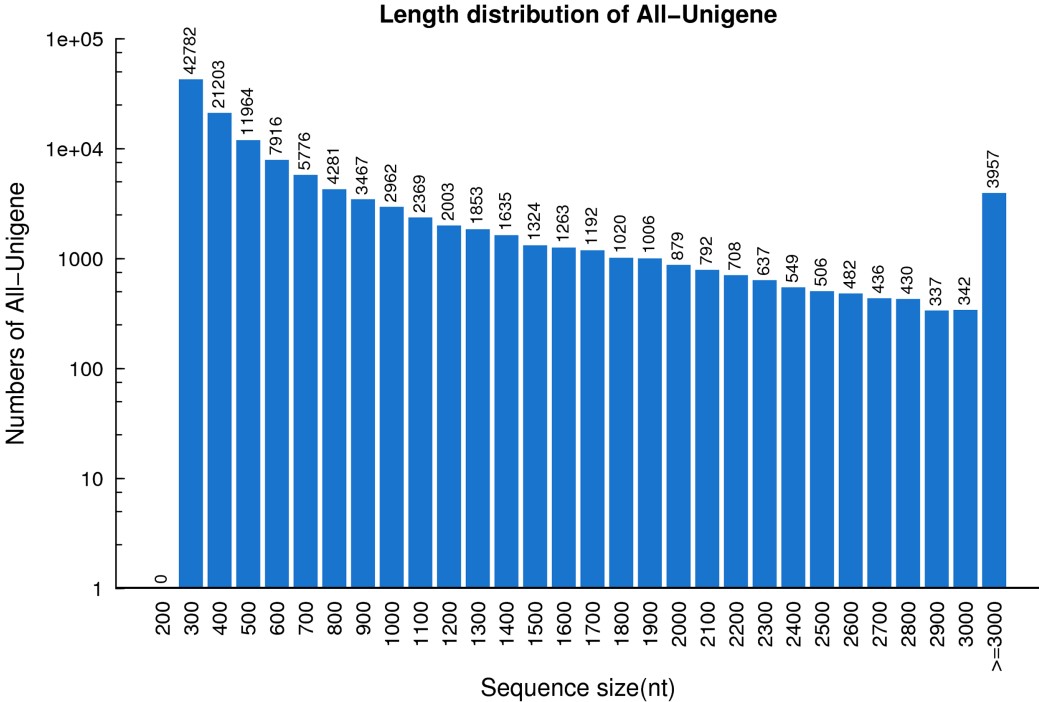

**Figure 1 The length distribution of the unigenes identified based on seven *R. arvalis* oviduct transcriptomes.** The *X*-axis shows the length distribution (nt) of sequenced unigenes and *Y*-axis indicates number of unigenes for a given length.

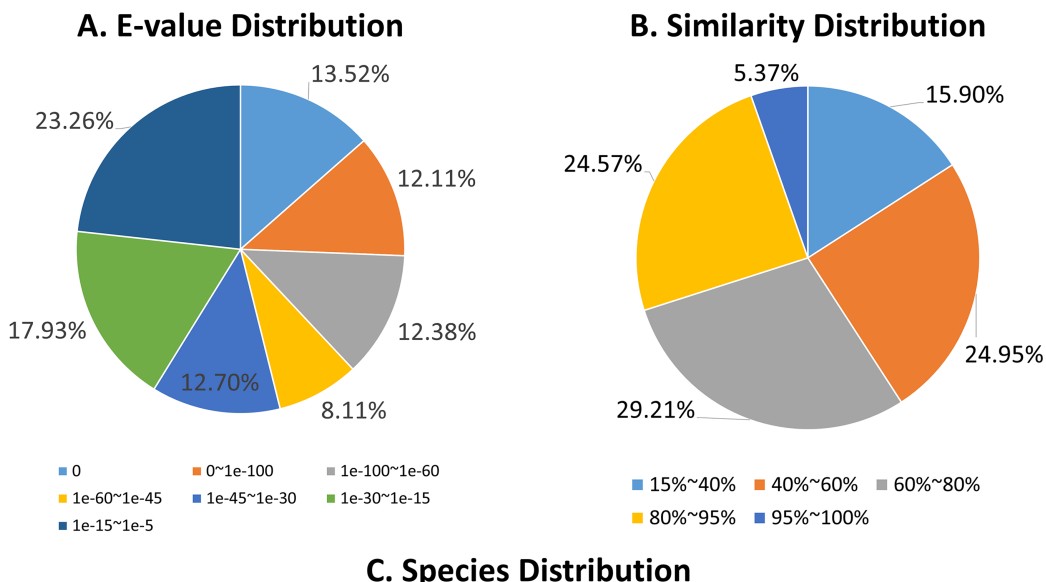

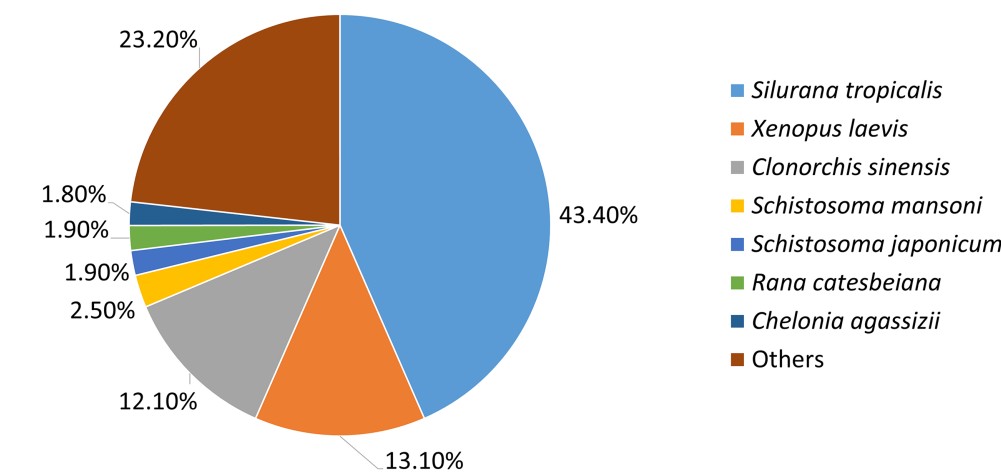

**Figure 2** **Annotation of *R. arvalis* unigenes against the Nr database.** (A) *E*-value distribution of the top BLAST hits for each unique sequence. (B) Similarity distribution of the top BLAST hits for each unique sequence. (C) Species distribution of the top BLAST hits for all homologous sequences.

*S. mansoni*: 72.3%; *S. japonicum*: 58.0%, respectively), indicating that we indeed sequenced parasites in the oviduct of these wild collected moor frogs. This warrants some caution in further analyses of our *R. arvalis* oviduct transcriptomes.

Functional classes were successfully annotated for 13,501 unigenes using COG (Figs. 3 and 4). BLASTX against Swiss-Prot, KEGG, NT, and GO database resulted in 37,262, 31,405, 39,138, and 24,452 hits, respectively (Fig. 4). Altogether, 57,839 unigenes (46.6% of all 124,071 unigenes) had significant matches with existing databases (Fig. 4). Among the 124,071 unigenes, 48,850 (39.4%) were predicted as protein coding sequences (CDS). Of these, 44,809 unigenes were aligned using existing databases, while 4,041 unigenes that could not be annotated with any database were predicted by EST Scan. The length distribution of the CDS protein sequences is available in Supplementary Information S1.

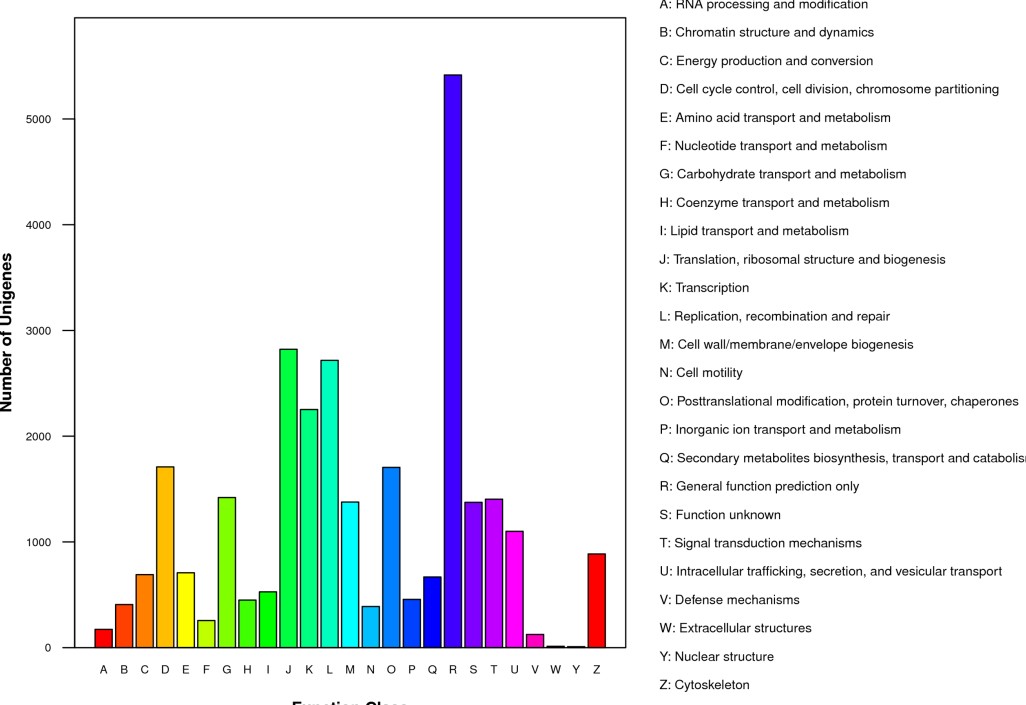

**Figure 3 COG functional classification of unigenes identified from the *R. arvalis* oviduct unigenes.**
The *X*-axis shows the different functional classes, and *Y*-axis the number of genes annotated into a given class. Most genes are in the classes of "General function," followed by "Translation, ribosomal structure, and biogenesis," "Transcription," and "Replication, recombination, repair."

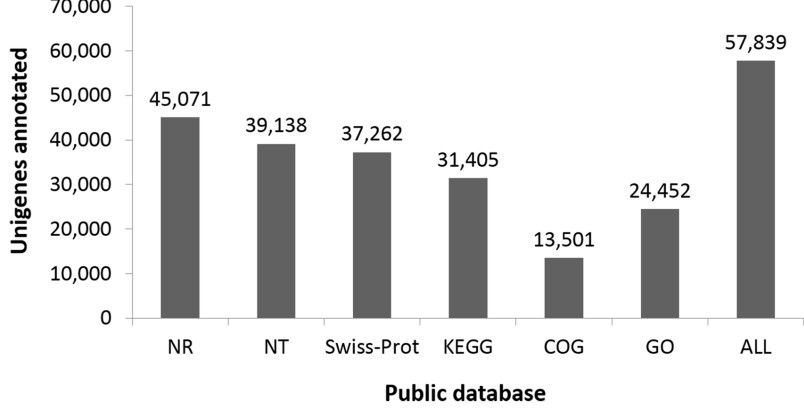

**Figure 4 Number of unigenes annotated based on different public databases (see 'Methods' for details on databases).**

## Functional pathway annotation

A total of 24,452 unigenes (42.27% of all annotated unigenes) were successfully categorized into 60 GO functional groups (Fig. 5). These were classified to three major categories: biological processes (23), cellular components (18), and molecular functions (19) (Fig. 5).

**All-Unigene GO Classification**

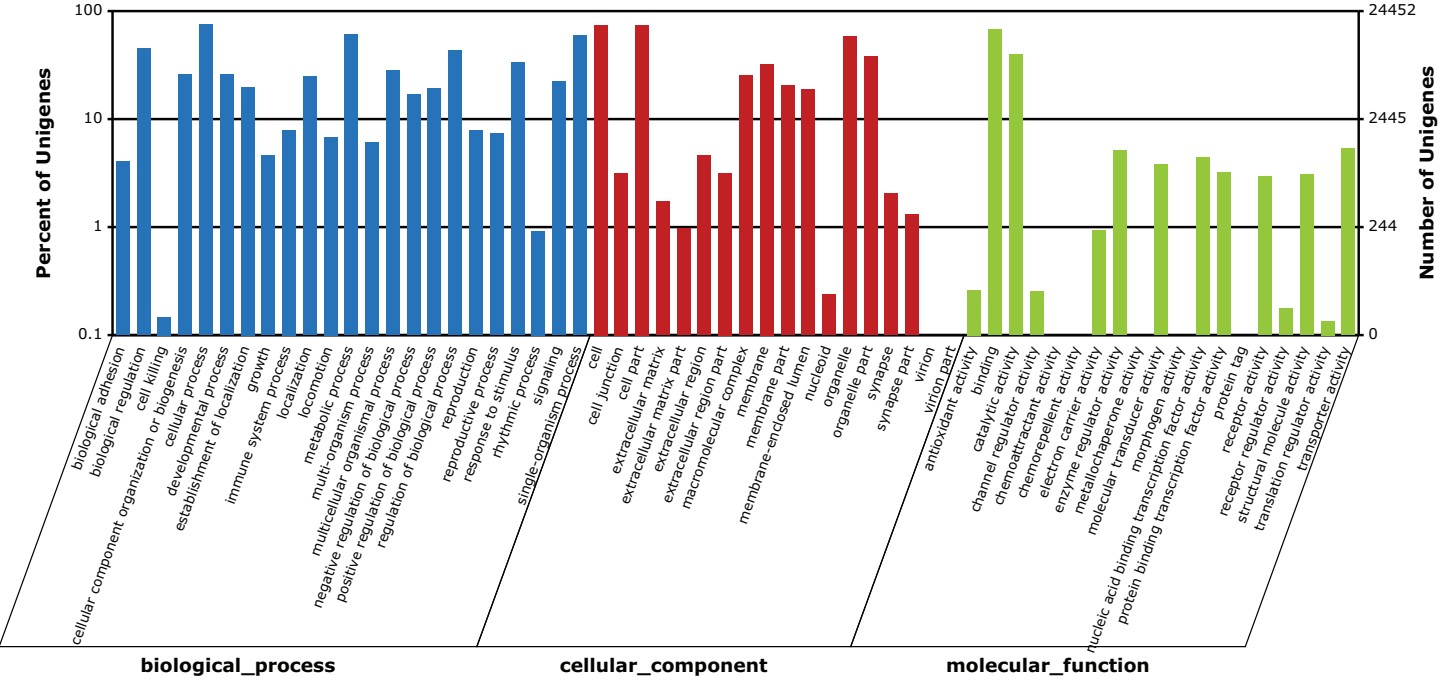

**Figure 5 GO categories of unigenes identified from the transcriptome of seven *R. arvalis* oviduct samples.** The unigenes were annotated in three categories as represented on the *X*-axis: biological processes (23), cellular components (18), and molecular functions (19). The *X*-axis indicates the GO term, while the *Y*-axis (log scale) indicates the number and percentage of unigenes for each GO term.

A total of 31,405 unigenes (54.29% of all annotated unigenes) were annotated in KEGG, and assigned to 259 known KEGG pathways (Table S2). The highly enriched pathways included: metabolic pathways (3,796, 12.09%), purine metabolism (1,910, 6.08%), regulation of actin cytoskeleton (1,202, 3.83%), focal adhesion (1,161, 3.7%), and calcium signaling pathway (1,159, 3.69%).

## Candidate genes for egg jelly coat

We identified two groups of candidate genes likely related to egg jelly coats: (i) core protein genes (Table 3) and (ii) protein glycosylation genes (Table 4). The major components of egg jelly core protein (ECM fiber) are mucins (*Hedrick & Hardy, 1991*), of which 13 and 11 different types, respectively, were detected in our dataset (Table 3). Within the Mucin gene family, Mucin-5AC, Mucin-5B, and Mucin-2 were very highly expressed (Table 3) and the most abundant transcripts of all unigenes. Previous evidence suggests that the Mucin-5 gene is expressed exclusively in the *X. tropicalis* oviduct, making Mucin-5 genes likely candidates for egg jelly coat genes (*Lang et al., 2016*). Several other minor components were also identified (Dermatopontin, Fibulin-5 and Fibrinogen-like protein 1, Decorin, EMILIN-1, Fibrillin-1, Fibronectin, and Laminin) (Table 3).

In addition to the candidate core protein genes, five major biosynthesis pathways involved in protein glycosylation were identified based on KEGG: Mucin type O-glycans, Other types of O-glycans, Heparan sulfate, Chondroitin sulfate, and Keratan sulfate

**Table 3 Genes coding for core proteins as identified from the oviduct of six *R. arvalis* females. Genes highlighted in bold are the most highly expressed core glycoprotein genes. See text for detailed discussion.**

| Component | Gene | Function |
|---|---|---|
| Mucin | Mucin-1 | ECM protein |
| | **Mucin-2** | ECM protein |
| | Mucin-4 | ECM protein |
| | **Mucin-5AC** | ECM protein |
| | **Mucin-5B** | ECM protein |
| | Mucin-6 | ECM protein |
| | Mucin-7 | ECM protein |
| | Mucin-15 | ECM protein |
| Collagen | Collagen alpha-1(I) | ECM protein |
| | Collagen alpha-1(III) | ECM protein |
| | Collagen alpha-1(V) | ECM protein |
| | Collagen alpha-1(XI) | ECM protein |
| | Collagen alpha-1(XII) | ECM protein |
| | Collagen alpha-1(XVIII) | ECM protein |
| | Collagen alpha-1(XXVII) | ECM protein |
| | Collagen alpha-2(I) chain | ECM protein |
| | Collagen alpha-2(IV) | ECM protein |
| | Collagen alpha-2(V) | ECM protein |
| | Collagen alpha-2(VI) | ECM protein |
| | Collagen alpha-5(IV) | ECM protein |
| | Collagen alpha-6(IV) chain | ECM protein |
| Others | Decorin | ECM protein |
| | Dermatopontin | ECM protein |
| | EMILIN-1 | ECM protein |
| | EMILIN-2 | ECM protein |
| | Fibrillin-1 | ECM protein |
| | Fibrinogen-like protein 1 | ECM protein |
| | Fibronectin | ECM protein |
| | Fibulin | ECM protein |
| | Laminin | ECM protein |

(Table 4). Of these, Mucin type O-glycan genes are the most likely candidates for egg jelly coat glycosylation (*Coppin et al., 1999*; *Lang et al., 2016*) (Fig. 6). Of the eight types of Mucin type O-glycans, only genes related to cores 1, 2, 3, 4, and 6 of Mucin type O-glycan biosynthesis were detected, while cores 5, 7, and 8 were not expressed in *R. arvalis* (Fig. 6).

## Preliminary differential expression analysis

Differentially expressed genes were identified by pairwise comparisons between each population pair ([S] vs [T], [S] vs [B], [T] vs [B]) of six of the females (the not ovulated female was excluded from these analyses). Overall, 4,457, 4,198, and 3,691 DEGs were identified between [S] vs [T], [S] vs [B], and [T] vs [B], respectively (Fig. S3). In general,

**Table 4 Protein glycosylation genes as identified from the oviduct of *R. arvalis* females.**

| Glycan pathway | Glycan type | Gene |
|---|---|---|
| Mucin type O-glycan | | beta-1,3-*N*-acetylglucosaminyltransferase |
| | | alpha-*N*-acetylgalactosaminide alpha-2,6-sialyltransferase |
| | | beta-1,6-*N*-acetylglucosaminyltransferase |
| | | beta-1,4-galactosyltransferase 5 |
| | | glycoprotein-*N*-acetylgalactosamine 3-beta-galactosyltransferase |
| | | *N*-acetylglucosaminyltransferase 3, mucin type |
| | | polypeptide *N*-acetylgalactosaminyltransferase |
| | | sialyltransferase 4A |
| | | sialyltransferase 7A |
| Other type of O-glycan | O-linked GlcNAc type | Protein O-GlcNAc transferase |
| | O-linked Man type | beta-1,2-*N*-acetylglucosaminyltransferase |
| | | beta-1,4-galactosyltransferase 1 |
| | | carbohydrate 3-sulfotransferase 10 |
| | | dolichyl-phosphate-mannose-protein mannosyltransferase |
| | | glucuronosyltransferase |
| | | sialyltransferase 6 |
| | | 4-galactosyl-*N*-acetylglucosaminide 3-alpha-L-fucosyltransferase |
| | O-linked Fuc type | beta-1,4-galactosyltransferase 1 |
| | | peptide-*O*-fucosyltransferase |
| | | sialyltransferase 6 |
| | O-linked Glc type | protein glucosyltransferase |
| | | UDP-xylose:glucoside alpha-1,3-xylosyltransferase |
| | O-linked Gal type | collagen beta-1,*O*-galactosyltransferase |
| | | lysyl hydroxylase/galactosyltransferase/glucosyltransferase |
| Heparan sulfate | | alpha-1,4-*N*-acetylglucosaminyltransferase EXTL3 |
| | | alpha-1,4-*N*-acetylglucosaminyltransferase EXTL2 |
| | | glucuronyl/*N*-acetylglucosaminyl transferase EXT1 |
| | | *N*-deacetylase/*N*-sulfotransferase (heparan glucosaminyl) 2 |
| | | Heparan sulfate glucosamine 3-*O*-sulfotransferase 1 |
| Chondroitin sulfate | | chondroitin sulfate *N*-acetylgalactosaminyltransferase 1/2 |
| | | chondroitin sulfate synthase |
| | | galactosylxylosylprotein 3-beta-galactosyltransferase |
| | | galactosylgalactosylxylosylprotein 3-beta-glucuronosyltransferase 1 |
| | | protein xylosyltransferase |
| | | xylosylprotein 4-beta-galactosyltransferase |
| Keratan sulfate | | beta-1,4-galactosyltransferase 1 |
| | | beta-1,4-galactosyltransferase 4 |
| | | beta-1,3-*N*-acetylglucosaminyltransferase 7 |
| | | carbohydrate 6-sulfotransferase 2 |
| | | *N*-acetyllactosaminide beta-1,3-*N*-acetylglucosaminyltransferase |
| | | sialyltransferase 4A |

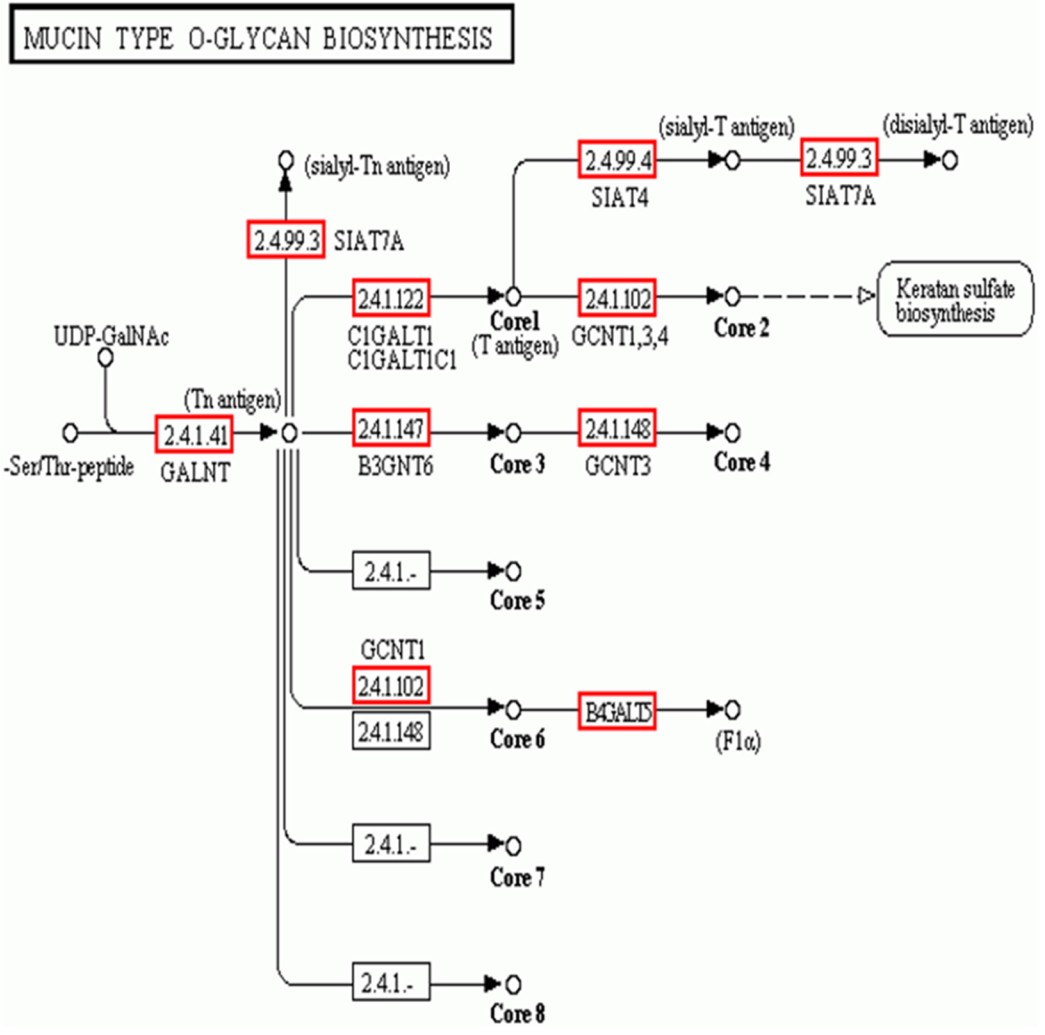

**Figure 6 The biosynthesis pathway (KEGG) of the Mucin type O-glycans.** Red squares indicate the genes expressed in the oviduct of *R. arvalis*. Study sites: GALNT, polypeptide *N*-acetylgalactosaminyltransferase; SIAT4, sialyltransferase 4A; SIAT7A, sialyltransferase 7A; C1GALT1, glycoprotein-*N*-acetylgalactosamine 3-beta-galactosyltransferase; GCNT1, beta-1,3-galactosyl-*O*-glycosyl-glycoprotein beta-1,6-*N*-acetylglucosaminyltransferase; B3GNT6, acetylgalactosaminyl-*O*-glycosyl-glycoprotein beta-1,3-*N*-acetylglucosaminyltransferase; GCNT3, *N*-acetylglucosaminyltransferase 3, mucin type; B4GALT5, beta-1,4-galactosyltransferase 5. 

[T] and [B] individuals had much lower gene expression levels compared to S individuals (Fig. S3). For instance, 3,397 and 3,388 genes were down-regulated in the [T] and [B] individuals compared to the [S] individuals, while only 1,060 and 810 genes were up-regulated in [T] individuals compared to the [S] individuals (Fig. S3). The smallest number of DEGs occurred between the [T] and [B] individuals, where 2,256 were up-regulated and 1,435 were down-regulated in [T] vs [B] individuals (Fig. S3).

 Enrichment of KEGG pathway in the differentially expressed unigenes was assessed using a Fisher's exact test (FDR <0.05). The FDR analyses identified 37 and 49 significantly enriched KEGG pathways in [T] (Table S4) and [B] (Table S3) compared to [S] individuals, whereas only two were identified between [T] and [B] (Table S5).
Again, the suggested expression profiles of [T] and [B] individuals were more divergent compared to that of [S] individuals, while differentiation between the [T] vs [B] was very small. The Ribosome (KO03010) and Oxidative phosphorylation (KO00190) were the most enriched pathways in both [S] vs [T] and [S] vs [B] comparisons (Tables S3 and S4). This indicated that [T] and [B] females had, in general, lower rates of energy production as well as protein biosynthesis.

Because we are able to rank each individual frog's offspring's degree of acid tolerance based on our previous work (Shu et al., 2016), we can test whether embryonic acid tolerance is related to the transcriptional activity in the mother's oviduct. The correlation analyses found 292 unigenes that were significantly correlated with acid tolerance. These genes have diverse functions ranging from signaling to protein biosynthesis, which could be suitable targets for future functional work (SI2). However, future studies are needed to confirm their functional correlations. The information of candidate genes was listed in Supplementary Information S2.

## SSRs and SNPs

A total of 26,711 SSRs were identified across all unigenes, in which Mono-nucleotide repeats (19,215), Di-nucleotide repeats (5,050), and Tri-nucleotide repeats (2,002) were the most abundant SSR motif classes, and Quad-nucleotide repeat (345), Penta-nucleotide repeats (79), and Hexa-nucleotide repeats (20) were detected at much lower frequencies (Fig. S1 and Table S1). A total of 231,274 SNPs were predicted across all unigenes. The number of SNPs in the seven individual cDNA libraries ranged from 63,354 to 86,608 (Fig. S2). The average Ts/Tv ratio (the numbers of transitions and transversions at the SNP sites) of all SNPs was ca. 1.75.

Due to a small sample size, the SSR and SNP data were not analyzed in detail for segregation among populations. However, as a first step we compared the homologous SNPs differentiated between population [S] (most acid sensitive) and population [T] (most acid tolerant) that differ in acid tolerance. We found 420 candidate SNPs that could be suitable targets for future functional work. The results are listed in Supplementary Information S3.

## DISCUSSION

Understanding the genetic architecture of maternal effects is important for our ability to understand the relative contribution of genetic vs. plastic effects in organismal phenotypes and the eco-evolutionary processes of natural populations (Mousseau & Fox, 1998; Räsänen & Kruuk, 2007). Yet given that maternal effects are typically strongly environmentally induced, and that the phenotypic variance of offspring is produced in the mother but their fitness consequences expressed in the offspring, establishing their genetic basis is challenging—particularly in non-model systems. We therefore applied RNA-seq analyses as a first step toward identifying candidate genes for maternal effects that are mediated through the gelatinous egg coats (i.e., egg jelly) in R. arvalis (Shu, Suter & Räsänen, 2015b). We next provide an overview of the general genomic variation revealed, followed by a discussion of putative candidate genes for maternal effects.

## General genomic aspects

We characterized 124,071 unigenes from the *R. arvalis* transcriptome, and successfully annotated 57,839 of them (46.6%). It is somewhat surprising that a substantial portion (ca. 15%) of the unigenes matched to a trematode parasite (Platyhelminthes) rather than any vertebrate genomes. Given that the *Xenopus* genomes are well annotated, it is possible that we accidentally have produced two transcriptomes: *R. arvalis* and trematode parasite (transcriptomics was recently used to detect amphibian endoparasites; *Santos et al., 2018*). Our additional analysis strongly indicates the presence of trematode parasites, but we can not currently identify the exact trematode species that presumably infect our frogs. Trematode parasites are common in amphibians (*Sears, Schlunk & Rohr, 2012*), but to our knowledge there are no previous reports on trematode parasites in amphibian oviducts and nothing is known of the oviduct parasites in *R. arvalis*. Hence, some caution is warranted when inferring the genes expressed in the oviducts in our study as part of the *R. arvalis* genome.

The lack of amphibian genomes is mostly due to their large genome sizes (due to large proportion of repeat sequences; *Sun et al., 2015*). Yet clearly more work on amphibian genomics is needed as (i) amphibians are the most ancient class of land-dwelling vertebrates and their genomic resources are essential for understanding vertebrate development and evolution; (ii) the understanding of evolutionary processes of amphibian populations would greatly be facilitated by studies on the genomic architecture of trait variation, and (iii) because conservation of amphibians that are under serious global decline (*Hof et al., 2011*; *Stuart et al., 2004*) could thus be facilitated (*Calboli et al., 2011*). The recently sequenced transcriptomes of *R. arvalis* (this study) and other Ranidae (*Birol et al., 2015*; *Helbing, 2012*; *Price et al., 2015*; *Qiao et al., 2013*; *Robertson & Cornman, 2014*; *Zhang et al., 2013*) will aid in developing amphibian genomics resources. As we used only oviduct tissue, while most other studies use other tissues (e.g., liver, skin), a comparison of oviduct-specific transcriptomes with transcriptomes from other tissues is needed to reliably identify oviduct-specific genes or expression profiles. For instance, when compared with the recently developed transcriptomic resources for the green frog *R. clamitans* and chorus frog *P. regilla* (*Robertson & Cornman, 2014*), the percentage of genes in our dataset that have hits in their dataset are 40.61% (20,402/50,238) and 40.62% (19,583/48,213), respectively, indicating a considerable proportion of candidate oviduct-specific genes. Furthermore, the large number of potentially amplifiable SSRs and SNP markers detected in our study (Supplementary Information) represent an important resource for applications in population genetics and for the detection of functional genetic variants (*Li et al., 2008*; *Morin, Luikart & Wayne, 2004*; *Schunter et al., 2014*).

## Candidate genes for maternal effects: inside the egg jelly coat

Although there is substantial potential for a genetic basis in maternal effects (*Räsänen & Kruuk, 2007*), most studies aiming to identify maternal effect genes have focused on their role in early embryonic development per se (*Tong et al., 2000*; *Wu et al., 2003*). To what extent maternal effect genes contribute to adaptive divergence of local populations and response to natural selection at early life stages is therefore still largely unknown.

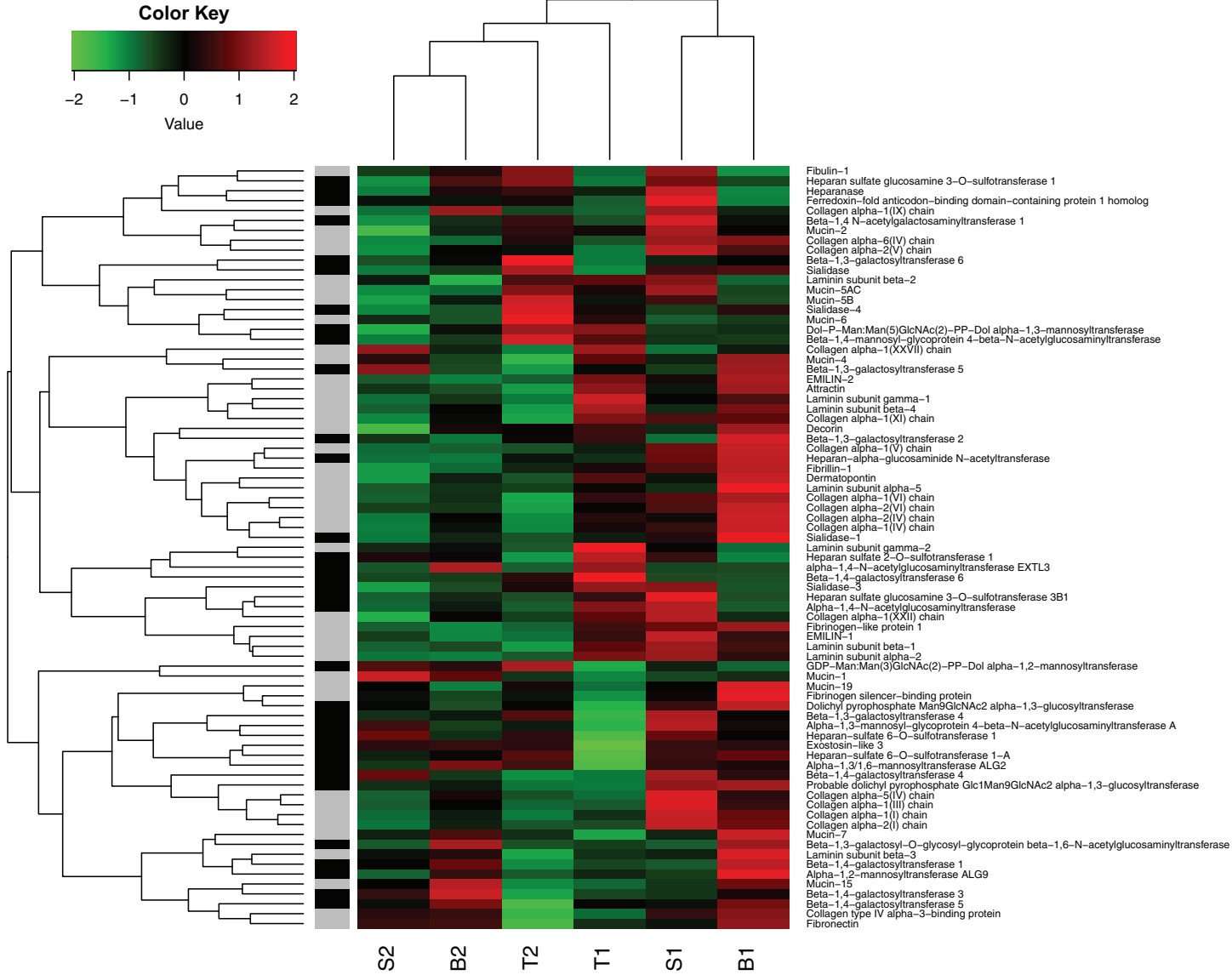

**Figure 7 Heat map of gene expression in oviducts of six *R. arvalis* females.** The genes presented are selected from those coding for core proteins (in gray) and protein glycosylation (in black). The colors represent high (red), low (green), or average (black) gene expression based on Z-score normalized FPKM values for each gene. The individual female's identity from the three study populations (T, S, and B) is indicated below. Within each population, the number indicates the individual female, whereby the females with the most sensitive embryos is indicated by 1 and female with the most acid tolerant embryos is indicated by 2 (acid tolerance was estimated in *Shu, Suter & Räsänen, 2015b*). B3 female was left out from this analysis because it had not fully ovulated at the time of sampling and hence was not directly comparable in gene expression patterns.

We identified two groups of candidate maternal effect genes based on their role in egg jelly biosynthesis: mucin core protein genes and protein glycosylation genes. The mucin genes and O-linked glycosylation genes are particularly likely candidates, given that amphibian egg jelly—including that of *R. arvalis* (*Coppin et al., 1999*)—mostly consists of mucin type O-linked glycoproteins (*Coppin et al., 1999*; *Guerardel et al., 2000*; *Strecker et al., 2003*). Evidence for high mucin gene expression in the oviduct has been found previously in *R. chensinensis* (*Zhang et al., 2013*) and *X. tropicalis* (*Lang et al., 2016*).

Whereas Mucin 2 is expressed in various tissues, Mucin 5AC was reported to be exclusively expressed in the oviduct of *X. tropicalis*. Although Mucins, particularly Mucin 5, are likely candidates for the core protein in jelly coats, the role of different genes contributing to the glycosylation of the jelly coat proteins are much more difficult to infer. For instance, the egg jelly coat of *R. arvalis* consists of at least 19 different glycan building blocks (*Coppin et al., 1999*) and our own analyses indicate high among individual polymorphism in the macromolecular composition of *R. arvalis* jelly overall (*Shu et al., 2016*) as well as in glycans. Given the complexity of O-linked glycosylation it is not surprising that multiple glycosylation genes are highly expressed in the *R. arvalis* oviduct—and the genetic basis of jelly coat mediated adaptive maternal effects could be complex.

In general, the genes representing likely candidate genes for egg jelly coat formation did show a different expression pattern compared to the global profile (Fig. 7). For instance, Mucin-5AC and Mucin-5B (i.e., the major components of jelly core proteins), were relatively highly expressed in the [T] individuals. However, expression of the putative jelly coat genes was very diverse across the six individuals in our data set (Fig. 7). It is important to note that the current data only provide the first step. More detailed analyses on coding vs regulatory variation underlying this heterogeneity, to what extent variation is due to coding and/or transcriptional differences in the maternally influenced jelly coat (which, in turn, affects embryonic survival), as well as SNP genotyping of allelic variation across the acidification gradient are important future avenues of study.

The potential complexity of the jelly phenotype and its function is also highlighted in the high degree of variation in expression of the putative jelly coat genes across different individuals in our study (Fig. 7). Given the many important roles that the egg jelly coats play in sperm–egg interactions, pathogen defense, and responses to various environmental stressors (reviewed in, *Menkhorst & Selwood, 2008*; *Shu, Suter & Räsänen, 2015b*), this complexity is not surprising. Follow-up work establishing the links between the genotype (jelly coat coding genes and variation in the expression)—phenotype (jelly coat glycome)—fitness (embryonic acid tolerance) map needed to confirm the role of the candidate genes is ongoing in our lab.

## CONCLUSIONS

In conclusion, we identified several mucin and O-linked glycosylation genes that are highly expressed in the oviduct of *R. arvalis*—and show high heterogeneity in expression. Given the role of *R. arvalis* in a broad range of evolutionary ecological questions, we believe this transcriptomic dataset together with the predicted SSR and SNP markers (Supplementary Information) will benefit future studies of molecular ecology and evolution in natural populations. We further hope that our oviduct transcriptomes lay the ground for future studies on the molecular evolution of jelly coat genes, thereby contributing to the emerging field of glycobiology in evolutionary ecology (*Shu, Suter & Räsänen, 2015b*; *Springer & Gagneux, 2016*), as well as studies on how these genes contribute to responses to natural selection at early life stages and adaptive divergence of local populations in particular. In particular, we hope that the genes identified here will aid in disentangling the genetic architecture of egg coat evolution and adaptive maternal effects.

## ACKNOWLEDGEMENTS

We thank Anssi Laurila for generous access to research facilities and Beatrice Lindgren for invaluable help in the field and laboratory. We thank particularly Ellen Menkhorst, Daniel Jeffries, Alice Dennis, Tyler Larsen, and an anonymous reviewer for valuable comments on earlier drafts of this manuscript as well as the Genetic Diversity Center (GDC) of ETH-Zurich for advice on genetic analyses.

### Funding

This study was supported by the Swiss National Science Foundation (to Katja Räsänen). Longfei Shu is currently a Simons Foundation Fellow of the Life Sciences Research Foundation. The funders had no role in study design, data collection and analysis, decision to publish, or preparation of the manuscript.

### Grant Disclosures

The following grant information was disclosed by the authors:
Swiss National Science Foundation.
Simons Foundation Fellow of the Life Sciences Research Foundation.

### Competing Interests

The authors declare that they have no competing interests.

### Author Contributions

- Longfei Shu conceived and designed the experiments, performed the experiments, analyzed the data, analysis tools, prepared figures and/or tables, authored or reviewed drafts of the paper, approved the final draft.
- Jie Qiu analyzed data, contributed analysis tools, prepared figures and/or tables, authored or reviewed drafts of the paper, approved the final draft.
- Katja Räsänen conceived and designed the experiments, contributed reagents/materials/analysis tools, authored or reviewed drafts of the paper, approved the final draft.

### Animal Ethics

The following information was supplied relating to ethical approvals (i.e., approving body and any reference numbers):

The experiments were conducted under permissions from the Västra Götaland county board (Dnr 522-6666-2011) and the Ethical committee (Dnr C65/11) for animal experiments in Uppsala County.

### Data Availability

Shu, Longfei (2018): Moor frog transcriptome. figshare. Fileset. https://doi.org/10.6084/m9.figshare.6025301.v1

## Supplemental Information

Supplemental information for this article can be found online at http://dx.doi.org/10.7717/peerj.5452#supplemental-information.

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
