# Peer review of "De novo oviduct transcriptome of the moor frog Rana arvalis: a quest for maternal effect candidate genes"

_PeerJ, doi:10.7717/peerj.5452_

## Round 0.1 · original submission · Minor Revisions

There are three excellent reviews of your ms, all of which recommend minor revisions, a decision I sanction. Reviewer 1 points out that you need to check all taxonomy reported in the ms. I suggest checking every amphibian name in Frost (AMNH online database) to ensure that your taxonomy is up to date. Reviewer 1 also suggests a lot more literature that you may want to consider. In addition to the extra genomes mentioned, that of X. laevis was published in 2016 (Session et al Nature, 538(7625), p.336). Reviewer 2 also has some excellent comments and suggestions for clarifying your ms. Reviewers 1 & 3 question the logic behind the parasite sequences, without having excluded for orthologous sequences from all vertebrates.

I look forward to receiving your resubmission.

Reviewer 1 ·

Basic reporting

In general, the quality of the writing and English is good and there is a sufficient amount of background material in the introduction as well as explanatory information in the materials and methods section. I note, though, that there are some sentence or grammar issues throughout – these aren’t grave but they are noticeable. For instance, on line 360 “high among individual polymorphism…is high” should have one fewer uses of “high”. The figures and tables are predominantly clear and easy to comprehend. Though in Figure 2C, there are a few issues with the species labeling. The second species is given only a common name (African clawed frog) whereas all other taxa are given their scientific names – this should be Xenopus laevis. Additionally, the 3rd to last species is labeled Aquarana catesbeiana but no such species exists. This references the American bullfrog (Rana catesbeiana), a congener of the focus species of this manuscript separated by ~40my. “Aquarana” refers to a subgeneric clade of eastern North American frogs but is not the genus of this species. The liver fluke Clonorchis sinensis is misspelled as “siensis”. Also, for Figure 5, is the Y-axis on a log scale? But generally, the goal of the paper is laid out well and the analyses and discussion are pertinent to the results.

Experimental design

I find this work to be well within the scope of the journal. It is clear what the authors set out to do, their methods are generally easy to follow, and most importantly the work was generally done in a suitable way to accomplish the intended goal. Any clarifications I need are outlined below in the general comments.

Validity of the findings

My reading of this work as that the authors have done a sufficient job of producing, refining, and analyzing the data to identify candidate loci that might be involved with maternal effects of acid tolerance via manipulation the oocyte jelly coat.

Additional comments

• In your abstract and elsewhere you note that egg jelly coats are an important, often neglected source of maternal effects. I certainly agree. But perhaps it would be best to note that this is predominantly for external fertilizing species like most anamniotes (fishes and amphibians) and wouldn’t apply to most other tetrapods?
• I like the authors’ use of a gradient approach here, using three populations varying in acid tolerance as well as two individuals within each population varying in acid tolerance. I initially liked the idea of the unovulated control female but not sure she actually constitutes a real control without similar individuals in all three populations. Data from this individual are generally downplayed throughout this manuscript too and in large part that makes sense because this individual is really only expected to differ in transcript discovery and transcription levels but not architectural (i.e., coding) differences. However, given this unovulated individual is neglected from Figure 7, the key figure on transcription differences, perhaps remove most of the reference to this individual in the methods, results, and discussion and just point to her transcript information in the supplement and the eventual publication of her data on online databases?
• I wonder with the number of unigenes presented for three parasites (the Clonorchis and two Schistosoma), whether these are true R. arvalis transcripts or whether the authors acquired RNA from parasites. In Molecular Phylogenetics and Evolution, Juan Santos and colleagues recently mined transcriptome data to look at endoparasite diversity in the guts of Dendrobatid frogs (Diversity within diversity: Parasite species richness in poison frogs assessed by transcriptomics). I don’t know the answer, but perhaps the authors can discuss a bit whether these RNA seq data are the result of sequence similarity to parasite transcripts or whether these are actual parasitization of the reproductive tract. The authors touch on this very briefly (on line 323) but it’s probably worth discussing more given how prevalent it was in your results.
• If the authors were predicting amino acid sequences, could they also discuss whether these differences represent synonymous mutations or not? I think this would be relevant if we are talking about possible genomic architectural differences underlying “maternal effects”
• The authors mention a couple of times that only 2 amphibian genomes are available. This isn’t true. The bullfrog (Rana catesbeiana; draft genome in Hammond et al. 2017, Nature Communications), the Iberian ribbed newt (Pleurodeles waltl; genome in Elewa et al. 2017, Nature Communications), and the axolotl (Ambystoma mexicanum; in Nowoshilow et al. 2018, Nature).
• Out of curiosity, were there any hits to transcriptomic resources developed for the green frog (Rana clamitans) and chorus frog (Pseudacris regilla) published in Robertson and Cornman (2014, Molecular Ecology)? I imagine they would have showed up in your discussion if this was relevant but green frogs are relatively closely related to Rana arvalis so I thought it would be worth checking
• You mention a few times that most of the unigenes couldn’t be annotated with existing databases because a lack of genomics resources. But it also isn’t surprising given the authors only used a single tissue type (oviduct) and given most studies focus on other tissues (brain, heart, liver, kidney, skeletal muscle). The authors had a very good reason to target oviduct, but I’m not really surprised most of these genes weren’t annotatable.
• I think the authors need to spend some time defining their interpretation of genetic / genomic architecture. This term typically refers to coding differences and is relatable to the microsatellite repeat and SNP loci identified by the authors. However, general differences in the number of transcripts don’t relate to genomic architecture. These might instead relate to regulatory differences that could even be intronic and not looked at in this study. Maternal effects are generally considered a plastic response and therefore the authors can’t really get at that here given tissues were harvested from animals housed in similar conditions. What the transcription and architectural data show is that are innate differences both in sequences of functional regions of the genome as well as transcriptional differences among populations independent of current conditions, although transcriptional differences may also affected by the conditions experienced when these females were tadpoles. Basically, I think the authors need to be a bit more careful in discussing the material throughout this manuscript. As it reads now, it suggests that transcriptional and coding differences found here are related to maternal effects. What you’ve really identified are candidate loci that might be innately or plastically (or both) influence acid tolerance in different population. I believe the authors mean to say this, and the title does indeed signify a quest for candidate loci, but the manuscript reads as if it is suggesting the differences detected here are driving maternal effects.
• Lines 278-286 discuss the number of transcriptional differences between populations. Because these are wild-caught adults, it is unclear whether differences are innate, genetic differences or whether they are plastic responses to tadpole rearing environment. Regardless, these are interesting results. The methods for these results aren’t particularly clear. Perhaps the authors could elaborate any statistical analyses done for this result and maybe provide a figure? Additionally, given the authors are able to rank each individual frog’s degree of acid tolerance from work elsewhere, perhaps the authors could use some regression analysis to test whether a frog’s acid tolerance is related to transcriptional activity at all loci or particular loci to draw some additional inference?
• Additionally, to what degree are various SSRs and SNPs related to each population’s degree of acidity? I fully recognize that the authors only analyzed 3 populations with only 2 individuals per population. But it would be tantalizing if there were loci where SSRs or SNPS segregated by the degree of acidity. This might possibly make these loci more suitable candidates for future functional / maternal effects work given you identified far more loci than you can possible assess in future analyses anyways.

·

Basic reporting

This manuscript is engaging and interesting to read and is written well.

With respect to the samples – ponds and individual animals, the method need to be more clear. Please be careful to define exactly what you mean by descriptive words eg. sensitive, mature.

Line 112 – maybe use as second set of brackets around the abbreviation for the study sites so it is clear that they are abbreviations eg. (Tottajarn [T].
Line 131 – this is not completely clear what you mean. Are these the most acid sensitive based on where there were captured, or the most sensitive from the ones captured at that site?
Line 232 – please define immature vs mature in the methods. Do you mean the ovulated vs not ovulated (line 133)? If so, immature/mature probably aren’t appropriate words.

Experimental design

Why is the pH in these ponds so different? Is it natural or is it from something else? Whatever is altering the pH in these ponds might alter the transcriptome of the frogs, it might not necessarily be the pH itself.

The hypothesis of this manuscript is that the frogs may alter their egg coat composition in order to cope with the changes in the environment seen at the 3 pond sites. Is there any evidence for this? Have the proteins contained within the egg jelly of this species been identified – how do the proteins identified relate to the genes you have identified? Can the egg coats themselves from the 3 ponds be analyzed?

Line 116 – is it really possible to say that S is the most acid sensitive and T is the most acid tolerant? Do you have breeding information from these sites – are these frogs able to breed as effectively in all sites or are they less able to breed in one site?

Line 134 – are there different regions within the frog oviduct as there are in mammalian, reptilian and avian oviducts? If there are, then differences in RNA transcription in specific regions of the oviduct may be missed by this method.

Validity of the findings

Overall, there is a lot of data presented in this manuscript, but due to the difficulties associated with working with this species, not a lot can be concluded at this point.

However the data presented is robust and appears well controlled (except for experimental questions above).

The questions being asked are interesting and worthy of continued research.

Additional comments

Thank you for the opportunity to review this interesting manuscript.

·

Basic reporting

Apart from a couple of typos (which I have included in the "Comments for the author" section) the manuscript is written in a clear, concise, self-contained and professional manner. Literature is well cited, data is well presented in figures.

Experimental design

This is a good first step in the process of identifying the desired genes. But I think a comparison of these oviduct-specific transcriptomes with transcriptomes from other tissues is needed in the future to find oviduct-specific genes or expression profiles. That said, this paper still represents a valuable resource.

Validity of the findings

no comment

Additional comments

Overall it is a solid paper - descriptive (necessarily so), but well executed and a useful resource.

Some more specific comments relating to the writing of the manuscript:

>>Introduction

104 - consider “encompassing” instead of “bracketing” here

108 - first set of transcriptomic markers, maybe, but not the first set of genome-wide markers. See this paper, where they generate RADseq markers across the genome:

Brelsford, A., Lavanchy, G., Sermier, R., Rausch, A. & Perrin, N. Identifying homomorphic sex chromosomes from wild-caught adults with limited genomic resources. Mol. Ecol. Resour. 17, 752–759 (2016).

Please amend the sentence to reflect this. Thanks.

>>Methods

185 - Kyoto Encyclopedia of Genes and Genomes (KEGG). You use KEGG earlier (167) - add the acronym definition there instead.

>> Results

No issues here

>> Discussion

320 - I think you mean liver fluke here.

319 - 324 - your suggestions for the fluke hits are as follows: i) due to poor annotation of amphibian genomes (I presume you are implying that the genes were therefore missed by the annotations of their genomes), ii) orthologous with genes of the liver fluke, or iii) R. arvalis may have been infested by other genetically related parasites.

I don’t quite agree with the priority given to these hypotheses, see below.

Re i): Given the quality of the Xenopus genomes, I would say that it is unlikely that so many genes would be missed by their annotation.
Re ii): The argument that these genes are orthologous to Platyhelminthes would also require that they are absent in the other existing amphibian (or any vertebrate) genomes for these to be the best hits for those genes. So that, to me is almost the same as i).
Re iii): For me this is the most likely explanation.

So, perhaps you could even just leave i) and ii) out, if the genes were absent from other amphibian genomes, I would not expect the next best hits to be Platyhelminthes, so I don’t think this result has occurred by accident. A useful thing to look at would be how much of the fluke genome these unigenes represent. It might be that you have actually produced two transcriptomes here, which would be cool.


384 - “lay” not “lays”

---

## Round 0.2 · Minor Revisions

Thank you for the revision of your manuscript. On reading it I have found some outstanding minor issues that need correction, and one larger issue that require your attention before I can accept your manuscript. Please make sure that you carefully read the entire manuscript for grammatical errors before resubmission. I have found a number of issues especially in the new text.

Additional analyses

L274 – Additional analyses: To be fair, I think that these would be better phrased as Post-hoc analyses, in order to make it clear that you carried them out after your initial study was conducted. Moreover, you should not report results of these additional analyses in the discussion (L413-415).
I can’t find mention of which individuals you found flukes in. It would be unusual to have the same parasite in every individual sampled from 3 populations (unless contamination came during your common garden experiment).
L278 – “To test how much of the fluke genome…” This should be in a separate paragraph as it is unrelated to the analysis on embryonic acid tolerance. Also, it is not clear whether there are one or several fluke species in your results. Your language needs to reflect this. See also L411 “…of two taxa…”
L286 – “…used them as query…” grammar, and elsewhere in this paragraph there are corrections needed.
L368 - "...note that to really validate..." grammar
L369 - "The information of candidate genes were listed..." grammar
L380 - "...as sample size is small..." grammar
L412-L413 is a repeat of L278.
L401-403 – this is a repeat of the text at L303-305.
L408 – “It is at the first look…” grammar
L415 - "...we indeed likely sequences..." grammar
L418 – The further studies statement is vacuous. The finding that you report is that there were trematodes, and I don’t think that any more studies are needed to confirm this.
L421-L425 – this is a repeat of the text at L105-L109.
L484 – “…that in future allows to test…” grammar
L119 – Ranid = ranid

---

## Round 0.3 · Minor Revisions

Thanks for your revision. Unfortunately, once again your revised text contains numerous grammatical errors. It is your responsibility to find sufficient help to correct these.

---

## Round 0.4 · accepted · Accept

Thanks for making these changes.

#